# The Outcome Relevance of Pre-ECMO Liver Impairment in Adults with Acute Respiratory Distress Syndrome

**DOI:** 10.3390/jcm12144860

**Published:** 2023-07-24

**Authors:** Stany Sandrio, Manfred Thiel, Joerg Krebs

**Affiliations:** Department of Anesthesiology and Critical Care Medicine, University Medical Centre Mannheim, Medical Faculty Mannheim, University of Heidelberg, Theodor-Kutzer-Ufer 1-3, 68165 Mannheim, Germany; manfred.thiel@medma.uni-heidelberg.de (M.T.); joerg.krebs@umm.de (J.K.)

**Keywords:** ARDS, ECMO, liver injury, MELD score

## Abstract

We hypothesize that (1) a significant pre-ECMO liver impairment, which is evident in the presence of pre-ECMO acute liver injury and a higher pre-ECMO MELD (model for end-stage liver disease) score, is associated with increased mortality; and (2) the requirement of veno-veno-arterial (V-VA) ECMO support is linked to a higher prevalence of pre-ECMO acute liver injury, a higher pre-ECMO MELD score, and increased mortality. We analyze 187 ECMO runs (42 V-VA and 145 veno-venous (V-V) ECMO) between January 2017 and December 2020. The SAPS II score is calculated at ICU admission; hepatic function and MELD score are assessed at ECMO initiation (pre-ECMO) and during the first five days on ECMO. SOFA, PRESERVE and RESP scores are calculated at ECMO initiation. Pre-ECMO cardiac failure, acute liver injury, ECMO type, SAPS II and MELD, SOFA, PRESERVE, and RESP scores are associated with mortality. However, only the pre-ECMO MELD score independently predicts mortality (*p* = 0.04). In patients with a pre-ECMO MELD score > 16, V-VA ECMO is associated with a higher mortality risk (*p* = 0.0003). The requirement of V-VA ECMO is associated with the development of acute liver injury during ECMO support, a higher pre-ECMO MELD score, and increased mortality.

## 1. Introduction

Extrapulmonary organ dysfunction has been associated with poor outcomes in patients with acute respiratory distress syndrome (ARDS) managed with extracorporeal membrane oxygenation (ECMO). A meta-analysis of two randomized controlled trials (conventional ventilator support versus extracorporeal membrane oxygenation for severe acute respiratory failure “CESAR” and ECMO to rescue lung injury in severe ARDS “EOLIA”) suggested that veno-venous (V-V) ECMO lacks the ability to improve the outcome of ARDS patients with more than two organ failures [1] and that mortality in patients receiving ECMO for respiratory failure is correlated with the amount and the extent of extrapulmonary organ dysfunction at the time of ECMO initiation [2]. In these patients, cardiovascular failure due to shock and sepsis contributes disproportionately to mortality [3,4,5,6]. Additionally, hepatic dysfunction, which is known to be an independent factor contributing to mortality in ARDS [7], might play a role in determining the outcome of respiratory ECMO [8,9]. To date, studies assessing the outcome relevance of liver dysfunction and injury before the initiation of ECMO support have centered on patients supported by veno-arterial ECMO for cardiogenic shock [10,11]. Given the limited literature, evaluation of the relevance of liver injury and dysfunction before and after the initiation of ECMO therapy in relation to the outcomes of ARDS patients supported with ECMO is warranted.

Liver dysfunction refers to impaired clearance and synthetic hepatic function with increased bilirubin and international normalized ratio (INR) [12]. Both values are incorporated in the Model for End-Stage Liver Disease (MELD) score, which has been proposed as a predictor of hepatic, cardiac, and renal dysfunction [13]. Among patients with liver failure, the MELD score has been shown to predict mortality [14]. The MELD score has been also reported as an outcome predictor in patients with respiratory or cardiocirculatory failure managed with V-V and veno-arterial ECMO [8,15].

Acute liver injury, also known as hypoxic liver injury, is diagnosed based on clinical criteria: (1) a massive, rapid, and often transient increase in serum transaminases, (2) the presence of a respiratory or cardiocirculatory failure with reduced hepatic oxygen delivery or utilization, and (3) the exclusion of other causes of liver injury, particularly drug- or viral-induced hepatitis [16,17]. Transaminases level at 2.5 to 20 times the normal upper limit has been used to define acute/hypoxic liver injury [17]. Henrion et al. reported that cardiac failure, particularly in conjunction with congestive heart failure, as well as respiratory failure and septic shock frequently causes acute liver injury [16]. Hence, ARDS patients with acute cor pulmonale due to elevated pulmonary artery pressure [18] or septic-induced vasoplegia that is unresponsive to catecholamines [19,20] might be especially vulnerable to acute liver injury due to systemic hypoxia, hepatic congestion, and diminished hepatic blood flow.

In ARDS with concomitant right ventricular failure due to acute cor pulmonale or septic cardiomyopathy, veno-veno-arterial (V-VA) ECMO might be indicated [3,21]. In this cannulation approach, the arterial outflow is bifurcated, with one portion directed retrograde towards the aorta and the other towards the right atrium [3,21,22,23]. This hybrid configuration combines the benefits and distinctive features of both V-V and veno-arterial ECMO, enabling concurrent and robust respiratory and circulatory support [21,22].

In this study, we hypothesize that in patients with primary respiratory failure:A significant liver impairment before ECMO initiation (pre-ECMO), indicated by the presence of acute liver injury or a higher MELD score, is associated with increased mortality;The requirement of V-VA ECMO support due to an acute cor pulmonale or cate-cholamine refractory shock is associated with (a) a higher prevalence of pre-ECMO acute liver injury and (b) a higher pre-ECMO MELD score and, therefore, (c) an increased mortality.

## 2. Materials and Methods

### 2.1. Data Acquisition, Inclusion, and Exclusion Criteria

After institutional ethics committee approval (Medizinische Ethikkommission II, University Medical Centre Mannheim, Medical Faculty Mannheim of the University of Heidelberg, study registration number 2021-881), and registration in the German Clinical Trials Register (DRKS00028509), a retrospective review of electronic medical records was performed to identify patients with V-V and V-VA ECMO support between January 2017 and December 2020 at the Department of Anesthesiology and Critical Care Medicine, University Medical Centre Mannheim, Germany.

We performed a comprehensive data collection for each eligible patient. We include all ARDS patients receiving V-V and V-VA ECMO due to primary respiratory failure. Patients who required ECMO support for other reasons (e.g., ECMO as intraprocedural support during aortic surgery, extracorporeal cardiopulmonary resuscitation) are excluded from the analysis. In these patients, we aggregate age, sex, body-mass index, diagnosis, duration of mechanical ventilation before ECMO initiation, the parameter of mechanical ventilation, the length and type of ECMO support, the length of ICU stay, the presence of chronic kidney or liver disease, the need of renal replacement therapy, history of pre-ECMO cardiac arrest, cardiac failure, septic shock, and central nervous system injury. We further collected laboratory data including the daily serum levels of total bilirubin, international normalized ratio (INR), aspartate aminotransferase (AST), and alanine aminotransferase (ALT). These comprehensive data points provide a detailed overview of the patients’ clinical profiles for further analysis.

### 2.2. ECMO Management

Our clinical workflow and management strategy for patients on ECMO support due to respiratory failure are detailed previously [3]. Briefly, in accordance with the EOLIA trial [5] and recent guidelines [24], V-V ECMO is initiated in severe hypoxic (PaO_2_/FiO_2_ < 80 for longer than six hours or PaO_2_/FiO_2_ < 50 for longer than three hours) or hypercapnic (arterial pH < 7.25 and PaCO_2_ > 60 mmHg for longer than six hours) ARDS patients [3,5,24]. V-VA ECMO is applied in patients with severe respiratory failure and concomitant hemodynamic instability with tissue hypoperfusion, a systolic blood pressure less than 90 mmHg, and a cardiac index less than 2.0 L/min/m^2^ despite preload optimization and the continuous infusion of catecholamines [3]. These patients commonly show primary respiratory failure accompanied by acute cor pulmonale or catecholamine-refractory septic shock.

Per the standard of our unit, we insert a 29 French multistage drainage cannula through the right femoral vein and a 23 French venous return cannula through the jugular vein [3]. In the case of V-VA ECMO, an additional 17 French arterial cannula and a 7 French antegrade perfusion cannula are inserted into the left femoral artery [3].

### 2.3. Definitions and Scores Calculation

In this study, acute liver injury is defined as the presence of increased serum aspartate transaminase greater than 350 U/L and alanine transaminase greater than 400 U/L, which indicated transaminase levels greater than 10 times the upper limit of normal [16,17]. Daily serum levels of transaminases, bilirubin, and creatinine and international normalized ratio (INR), and MELD score are assessed immediately prior to ECMO initiation (pre-ECMO) and during the first five days on ECMO support.

The MELD score is calculated according to the current Organ Procurement and Transplantation Network (OPTN) policies [25] and as recommended by the United Network for Organ Sharing [26,27].
$$\mathrm{MELD}=\left(0.957\times \ln\left(\mathrm{creatinine}\frac{\mathrm{mg}}{\mathrm{dL}}\right)+0.378\times \ln\left(\mathrm{bilirubin}\frac{\mathrm{mg}}{\mathrm{dL}}\right)+1.120\times \ln\left(\mathrm{INR}\right)+0.643\right)\times 10$$

In patients with serum creatinine above 4.0 mg/dL, as well as in patients who require a minimum of two dialyses or 24 h of continuous renal replacement therapy within the last seven days, the value for serum creatinine used in the calculation is set to 4.0 mg/dL [25]. For bilirubin or creatinine value less than 1 mg/dL, a value of 1 mg/dL is used in the calculation [25]. The MELD score is then rounded to the nearest integer and assessed at ICU admission, just before ECMO initiation (pre-ECMO) and during the first five days on ECMO support.

In this study, we analyze a patient cohort under V-V and V-VA ECMO for primary respiratory failure. Most of the patients presented with hypernatremia and, thus, the sodium value is set to 137 in the MELD-Na calculation [25]. This calculation resulted in identical MELD and MELD-Na values. Therefore, we use the MELD score in this study (not MELD-Na).

To further characterize the study population SAPS II (simplified acute physiology score II), SOFA (sequential organ failure assessment), RESP (respiratory ECMO survival prediction), and PRESERVE (predicting death for severe ARDS on V-V ECMO) scores are calculated. The SAPS II score is calculated as previously described by Le Gall et al. with physiological variables, which are collected within the first 24 h of treatment in the ICU [28]. SOFA, RESP, and PRESERVE scores are calculated at ECMO initiation, as described by Vincent et al. and Schmidt et al., respectively [29,30,31].

### 2.4. Statistical Analysis

Statistical analysis is performed with JMP^®^ Version 15 from SAS (SAS, Cary, NC, USA). Categorical variables are presented as frequencies of observation (%) and analyzed using a two-tailed Fisher’s exact test. Continuous variables are reported as medians with corresponding 25–75% interquartile ranges and comparisons are made using the Wilcoxon nonparametric test. For data that are measured multiple times, a repeated measures ANOVA and F-test are employed for analysis.

The following risk factors are included in our analysis: age, sex, body-mass index, ECMO type (V-V or V-VA), relevant comorbidities (pre-ECMO cardiac failure, septic shock, preexisting chronic liver and renal diseases, as well as the presence of acute liver injury), SAPS II at ICU admission, as well as pre-ECMO MELD, SOFA, PRESERVE, and RESP scores. The ability of a risk factor to predict mortality is assessed with logistic regression. The cut-off values of a risk factor for predicting mortality are correspondingly determined through a ROC curve analysis.

As we aimed to evaluate the impact of extrapulmonary organ function at the time of ECMO initiation on mortality, univariate and multivariable analyses are based on values at ECMO initiation (pre-ECMO). The multivariable analysis includes all factors with a *p* ≤ 0.05 at the univariate analysis. To avoid redundancy, single laboratory values (i.e., bilirubin, creatinine, INR, aspartate, and alanine transaminase) are excluded from the analysis.

The links between the requirement of V-VA ECMO support and (1) a higher prevalence of pre-ECMO acute liver injury, (2) a higher pre-ECMO MELD score, and (3) increased mortality are evaluated with logistic regression.

Survival estimates are completed with Kaplan–Meier and Cox proportional hazards analyses. Patients who were discharged alive from ICU are censored at the time of their discharge date.

## 3. Results

Between January 2017 and December 2020, we identified 187 ECMO runs (42 V-VA and 145 V-V ECMO) on 177 patients. Eight patients required two ECMO runs and one patient required three ECMO runs due to recurring respiratory failure.

### 3.1. Patient’s Demographics and Characteristics

The patients’ demographics and characteristics are presented in Table 1. Survivors have significantly lower SAPS II (69 (59–80) vs. 78 (64–90), *p* = 0.002), a lower incidence of cardiac failure (19% vs. 43%, *p* = 0.0005), and require significantly less V-VA ECMO support (12% vs. 34%, *p* = 0.0004). There is no significant difference in the pre-ECMO prevalence of septic shock and preexisting chronic liver or renal diseases between survivors and nonsurvivors. However, survivors show a lower pre-ECMO prevalence of acute liver injury (*p* = 0.03) and have a lower MELD score (12 (8–20) vs. 19 (11–23), *p* = 0.0004), SOFA score (13 (11–16) vs. 15 (13–17.7), *p* = 0.001), PRESERVE score (3 (2–5) vs. 4 (3–6), *p* = 0.005), and RESP score (1 (2–3) vs. 0 (−2–2), *p* = 0.04).

Survivors also show a significant lower pre-ECMO bilirubin (0.6 (0.3–1.2) vs. 0.9 (0.5–1.8), *p* = 0.01), creatinine (1.4 (0.7–2.4) vs. 1.8 (1.1–2.9), *p* = 0.01), INR (1.1 (1.0–1.2) vs. 1.2 (1.1–1.5), *p* < 0.0001), aspartate (77 (38.2–146.5) vs. 143 (58.2–414), *p* = 0.0002), and alanine transaminases (39 (28–70.2) vs. 53 (30–159), *p* = 0.02), Table 1.

### 3.2. The Development of Acute Liver Injury

Pre-ECMO acute liver injury is observed in 8 out of 145 V-V ECMO and 6 out of 42 V-VA ECMO cases (*p* = 0.09), Figure 1. Within the first five days after ECMO initiation, acute liver injury is identified in six additional patients on V-V ECMO and ten additional patients on V-VA ECMO (*p* < 0.0001), Figure 1.

### 3.3. The Course of MELD Score

Prior to ECMO initiation and during the first five days on ECMO, the repeated measures analyses show a significant increase of MELD score in both V-V ECMO (F test *p* < 0.0001) and V-VA ECMO (F test *p* = 0.005) groups, Figure 2. These are associated with increased total bilirubin and creatinine within individuals over time (F-test *p* < 0.0001 for both bilirubin and creatine). The increase in creatinine is contributed to the application of continuous renal replacement therapy and, thus, the creatinine value in the MELD score calculation is set to 4.0 mg/dL.

In the V-V ECMO but not the V-VA ECMO group, there is a significant difference in pre-ECMO MELD values between nonsurvivors and survivors (*p* = 0.01). However, there is a striking increase in MELD score in V-VA ECMO nonsurvivors as compared to the V-VA ECMO survivors.

### 3.4. Outcome Predictors

Table 2 outlines the ability of pre-ECMO risk factors (age, sex, body-mass index, pre-ECMO cardiac failure, septic shock, chronic liver and kidney diseases, acute liver injury, levels of bilirubin, creatinine, INR, and both transaminase enzymes, as well as SAPS II at ICU admission, MELD score, and ECMO type) to predict mortality.

In the univariate analysis, pre-ECMO cardiac failure, acute liver injury, bilirubin, transaminase enzymes, INR, pre-ECMO MELD, SOFA, PRESERVE and RESP scores, ECMO type, and SAPS II are related to ICU mortality. In the multivariable analysis, single laboratory values (i.e., bilirubin, creatinine, INR, aspartate, and alanine transaminase) are excluded from the analysis to avert redundancy. Here, only the pre-ECMO MELD score independently predicts ICU mortality (*p* = 0.04). The analysis shows a higher mortality in patients with a pre-ECMO MELD score greater than 16. Factors related to the pre-ECMO MELD score are summarized in Appendix A, Table A1.

### 3.5. The Impact of Liver Injury and a High Pre-ECMO MELD and SAPS II Scores on Outcome

According to the Cox proportional hazard model, acute liver injury occurring both before and after ECMO initiation is significantly associated with a 4.5-fold and 4.7-fold higher risk of mortality, respectively (*p* < 0.0001), Table 3. Additionally, the Cox model estimates a 1.9-fold and 2.3-fold higher mortality risk in patients with a pre-ECMO MELD score > 16 (*p* = 0.002) and SAPS II > 75 (*p* = 0.0001), Table 3.

Kaplan–Meier analyses reveal a notably higher survival probability within two months of ECMO initiation for patients who did not have pre-ECMO or developed acute liver injury during ECMO (Log-Rank *p* < 0.0001), Appendix A, Figure A1 and Figure A2.

Irrespectively of the ECMO strategies, the Kaplan–Meier analysis shows a worse 30 days survival probability for patients with an acute liver injury, both prior to ECMO initiation (log-rank *p* < 0.0001), Figure 3 and within the first five days of ECMO support (log-rank *p* < 0.0001), Appendix A, Figure A3.

In the Cox proportional hazard model, pre-ECMO acute liver injury is associated with a 5.4 higher mortality risk in the V-V ECMO group (*p* = 0.0001); while the higher mortality risk in V-VA ECMO groups is statistically nonsignificant (*p* = 0.07), Table 3. Among patients with pre-ECMO acute liver injury, both the V-V and V-VA ECMO groups show a similar mortality risk (95% CI 0.3–3.3, *p* = 1.0).

The incidence of acute liver injury within the initial five days of V-V and V-VA ECMO correlates with a 5.7-fold and 2.7-fold increase in mortality (*p* < 0.0001 and *p* = 0.01), respectively, Table 3. Among patients with acute liver injury during ECMO, the V-V ECMO group shows a higher mortality risk than the V-VA-ECMO group; however, this difference is statistically nonsignificant (95% CI 0.5–2.6, *p* = 0.7).

The pre-ECMO MELD score is significantly lower in the V-V ECMO group than in the V-VA ECMO group (13 (8–21) vs. 17 (13.5–25, *p* = 0.007). The Kaplan–Meier analysis shows a worse 30 days survival probability for patients with a pre-ECMO MELD score greater than 16 in both V-V and V-VA ECMO groups, Figure 4.

Among patients with a pre-ECMO MELD score > 16, mortality increases by 1.7 and 2.6 times in those receiving V-V and V-VA ECMO support, respectively (*p* = 0.04 and *p* = 0.0019, Table 3. When comparing the two ECMO strategies in patients with a pre-ECMO MELD score > 16, V-VA ECMO is associated with a 2.7 times higher mortality risk compared to V-V ECMO support (95% CI 1.6–4.7, *p* = 0.0003).

In patients with a pre-ECMO SAPS II > 75, mortality increases by 1.9 times for those on V-V ECMO support (*p* = 0.01) and 4 times for those on V-VA ECMO support (*p* = 0.0004), Table 3. Here, the V-VA ECMO group demonstrates a 3.2 times higher mortality risk than the V-V ECMO group (95% CI 1.9–5.6, *p* < 0.0001).

The univariate analyses show that the requirement of V-VA ECMO support is associated with the development of acute liver injury during ECMO support (*p* < 0.0001), a higher pre-ECMO MELD score (*p* = 0.01), and a higher ICU mortality (*p* = 0.0004). However, it is not linked to a higher prevalence of pre-ECMO acute liver injury (*p* = 0.09).

## 4. Discussion

This study’s main findings could be summarized as follows: (1) a significant pre-ECMO liver impairment, which is evident in the presence of pre-ECMO acute liver injury and a high pre-ECMO MELD score, is associated with increased mortality; (2) a pre-ECMO MELD score greater than 16 is an independent predictor of mortality in patients under ECMO support due to a primary respiratory failure; and (3) the requirement of V-VA ECMO support is associated with a higher pre-ECMO MELD score and increased mortality.

### 4.1. Acute Liver Injury

Our Cox analysis shows that the presence of pre-ECMO acute liver injury substantially increases the risk of ICU mortality. Hypoxic liver injury, also known as acute or ischemic liver injury, is characterized by a massive transaminases elevation resulting from reduced hepatic oxygen delivery or utilization [17]. Four mechanisms are potentially involved: (1) hypoxia, (2) ischemia due to hypoperfusion or hypotension, (3) hepatic venous congestion, and (4) the liver’s inability to extract and utilize oxygen [16,32]. Moreover, Seeto et al. suggested that liver hypoxia and ischemia resulting from low cardiac output are not alone sufficient to cause typical hypoxic hepatitis [33]. In their analysis, 94% of patients with acute liver injury had a right ventricular dysfunction and the accompanying hepatic venous congestion [33]. All mechanisms are commonly present in patients with ARDS and the associated septic shock or acute cor pulmonale, which reflects our patient cohort under V-V and V-VA ECMO support in this study.

In our institution, V-VA ECMO is typically initiated in ARDS with either acute cor pulmonale or catecholamine-refractory septic shock [3]. Prior to ECMO cannulation, these patients show a high illness severity and already exhibit multiorgan failure. As expected, the V-VA ECMO group shows a higher prevalence of acute liver injury prior to ECMO initiation (14%) and within the first five days on ECMO support (38%), as compared to the V-V ECMO group (5.5% and 9.5%, respectively), Figure 1. Hypoxia, hypotension, and venous congestion might be addressed with V-VA ECMO. However, V-VA ECMO cannot alleviate the liver’s inability to extract and utilize oxygen, which might occur in septic shock [16].

According to the findings presented in Table 3, pre-ECMO acute liver injury is associated with a significantly higher mortality risk in the V-V ECMO group. In the V-VA ECMO group, however, although the pre-ECMO transaminase levels are higher compared to the V-V group, the association between pre-ECMO acute liver injury and mortality does not reach statistical significance. This observation can be attributed to the profound hemodynamic instability in conjunction with hypoxemia prior to V-VA ECMO initiation, which contributes to mortality in V-VA patients irrespective of the presence or absence of pre-ECMO liver injury. As a result, the prognosis of patients receiving V-VA support is predominantly influenced by the severity of hemodynamic disturbance and the effectiveness of V-VA ECMO in rapidly stabilizing the cardio–circulatory system.

In this study, acute liver injury is defined as the presence of elevated serum aspartate transaminase levels exceeding 350 U/L and alanine transaminase levels surpassing 400 U/L. These thresholds, as suggested by Henrion et al., indicate transaminase levels that are more than 10 times higher than the upper limit of normal [16]. Both transaminase enzymes reach their peak levels within 24 h after a severe hemodynamic disturbance [17]. Given that the most severe hemodynamic disturbances typically occur during V-VA ECMO initiation [3], it is expected that both transaminase enzymes will reach their peak levels on the day following V-VA ECMO initiation.

Our results show that V-VA ECMO is linked to the occurrence of acute liver injury within the first five days of support (Table 3). However, among patients who develop an acute liver injury during ECMO, the V-VA ECMO group exhibits a lower mortality risk compared to the V-V ECMO group (Table 3). While this difference could be partially attributed to the ability of V-VA ECMO to stabilize hemodynamics, ensure adequate oxygen supply, and mitigate additional end-organ damage, the difference does not reach statistical significance (Table 3). Of note, our analyses include a relatively small sample size with only 42 V-VA ECMO runs. Consequently, the limited number of cases might not provide enough statistical power to establish a significant finding.

### 4.2. MELD Score as an Independent Outcome Predictor

MELD score is an objective metric and quickly assesses hepatic function [26]. It has a predictive value in acute liver failure [14] and has also been used widely to allocate livers for transplantation [26]. Wiesner et al. reported that without liver transplantation, patients with a MELD score ˂ 9 experienced a 1.9% mortality at three months, whereas patients with a MELD score ≥ 40 had a mortality rate of 71.3% [34].

Outside of liver cirrhosis and transplants allocation, the MELD score has been proposed as a predictor of renal, hepatic, and cardiac dysfunction [13]. As the outcome of respiratory ECMO, it is associated with nonpulmonary organ dysfunction at the time of ECMO initiation [2]; our results that demonstrate the pre-ECMO MELD score as an independent outcome predictor are in line with the findings from the CESAR and EOLIA trial [1]. In our analysis, the MELD score has a superior predictive performance for mortality compared to the SOFA, PRESERVE, RESP, and SAPS II scores (Table 2). In addition, the MELD score has been shown to have prognostic value in patients with respiratory failure supported by V-V ECMO and in patients with cardiac failure who required left ventricular assist devices [8,13]. As the MELD score calculation is based solely on three readily available, routinely collected, and reproducible laboratory values (creatinine, total bilirubin, and INR), it is easy to implement in a clinical setting and independent of subjective values [25].

Watanabe reported that a MELD score greater than 12 is an independent predictor of mortality in 71 patients with respiratory failure supported with V-V ECMO [8]. Our analysis of 42 V-VA and 145 V-V ECMO cases for primary respiratory failure also shows that the pre-ECMO MELD score is associated with mortality in both univariate (*p* = 0.0001) and multivariable (*p* = 0.04) analyses. The calculated cut-off value of 16 is slightly higher than previously reported by Watanabe et al.

A Cox proportional hazard analysis shows that a pre-ECMO MELD score greater than 16 increases the hazard ratio for ICU mortality by a factor of 1.9, Table 3. Severe ARDS is commonly associated with the progressive deterioration of nonpulmonary organ functions. This nonpulmonary organ dysfunctions, such as acute liver injury and dysfunction, coagulopathy, right heart dysfunction, catecholamine-refractory septic vasoplegia, or acute kidney failure, is reflected in the higher pre-ECMO MELD score and, therefore, might explain the value of the MELD score as an independent outcome predictor in patients with severe ARDS managed with ECMO.

In line with our findings, Matthews et al. reported the association between the MELD score prior to the implantation of ventricular assist devices and the postoperative right ventricular failure, renal failure, and mortality [13]. They reported that a preoperative MELD score greater than 17 is associated with a three-fold increased odds of perioperative mortality [13].

In contrast, Sern Lim reported a reduced predictive performance of the pre-ECMO MELD excluding the INR (MELD-XI) score in patients with acute decompensated chronic left heart failure bridged with veno-arterial ECMO [35]. These patients typically exhibit cardiac congestion and sympathetic and neurohormonal activation resulting in various degrees of hepatorenal impairment [35]. The author claims that the progressive multiorgan deterioration “homogenizes” his patient cohort and might thereby reduce the discriminatory value of the pre-ECMO MELD score [35]. In our study, however, we analyzed a rather homogenous patient cohort with severe ARDS and various degrees of extrapulmonary organ dysfunction. In this population, survival depends on the extent of extrapulmonary organ dysfunction at ECMO initiation [2], which is reflected by the pre-ECMO MELD score.

As a higher MELD score reflects a higher severity of illness with established organ dysfunction, our analysis shows a significantly higher pre-ECMO MELD score in the V-VA ECMO group than in the V-V ECMO group. V-VA ECMO is used to maintain hemodynamic stability in patients with respiratory failure and a concomitant acute right heart failure or catecholamine-refractory septic shock. As patients who are supported with V-VA ECMO already had a minimum of two failing organs (pulmonary and cardiovascular) prior to ECMO initiation, hepatorenal dysfunction, which is reflected in a higher pre-ECMO MELD score, further increases the mortality risk. The Cox model estimates that, among patients with a pre-ECMO MELD score greater than 16, V-VA ECMO support has a 2.7 times higher hazard ratio of ICU mortality, as compared to the V-V ECMO.

### 4.3. Limitations

Our analysis is subject to the limitations inherent in a retrospective study conducted at a single center, which includes the possibility of selection bias. The relatively small sample size, especially in the V-VA ECMO group, poses challenges in achieving robust comparability for statistical analysis. It is important to acknowledge these limitations when interpreting the findings and recognizing the potential impact they may have on the generalizability of the results.

## 5. Conclusions

A MELD score numerically operationalizes multiorgan dysfunction. A Pre-ECMO MELD score, both as continuous and as a dichotomous variable, is an independent outcome predictor in patients with primary respiratory failure supported with V-V or V-VA ECMO. Additionally, in our analysis, the MELD score has a superior predictive performance for mortality compared to the SOFA, PRESERVE, RESP, and SAPS II scores.

Immediately prior to V-VA ECMO initiation, patients are severely debilitated, experiencing multiorgan failure involving the lungs, heart, and vasomotor system. This condition typically arises due to acute cor pulmonale or catecholamine-refractory shock. The need for V-VA ECMO support to stabilize the pulmonary and cardio–circulatory systems is linked to a higher pre-ECMO MELD score and an elevated risk of mortality.

## Figures and Tables

**Figure 1 jcm-12-04860-f001:**
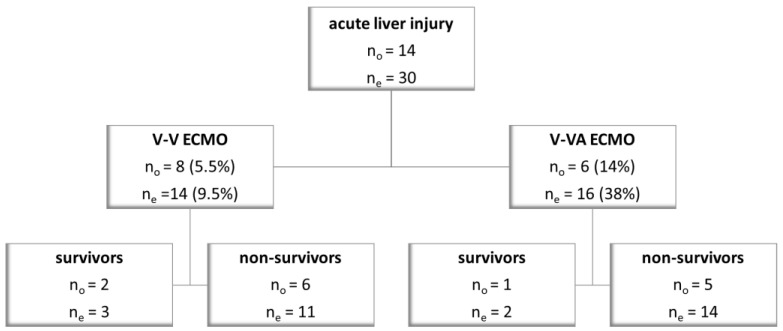
The prevalence of acute/hypoxic liver injury prior to ECMO initiation (n_o_) and within the first five days on ECMO support (n_e_).

**Figure 2 jcm-12-04860-f002:**
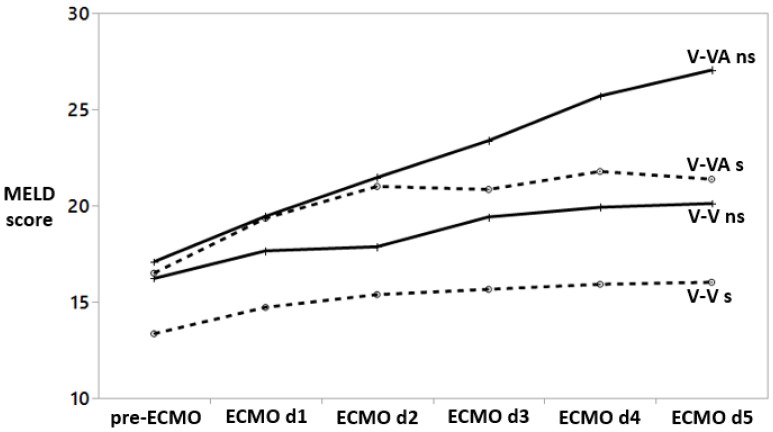
Repeated measures analyses of MELD scores at various assessment points. MELD values are displayed as means. Within individuals, MELD scores increase significantly over time both in V-VA ECMO (F-test *p* = 0.005) and in V-V ECMO groups (F-test *p* < 0.0001). However, in both V-V and V-VA groups, this increased MELD score does not significantly change the mortality rate over time (F-test *p* = 0.2 and *p* = 0.3, respectively). Dotted lines indicate survivors (V-V s and V-VA s); solid lines indicate nonsurvivors (V-V ns and V-VA ns).

**Figure 3 jcm-12-04860-f003:**
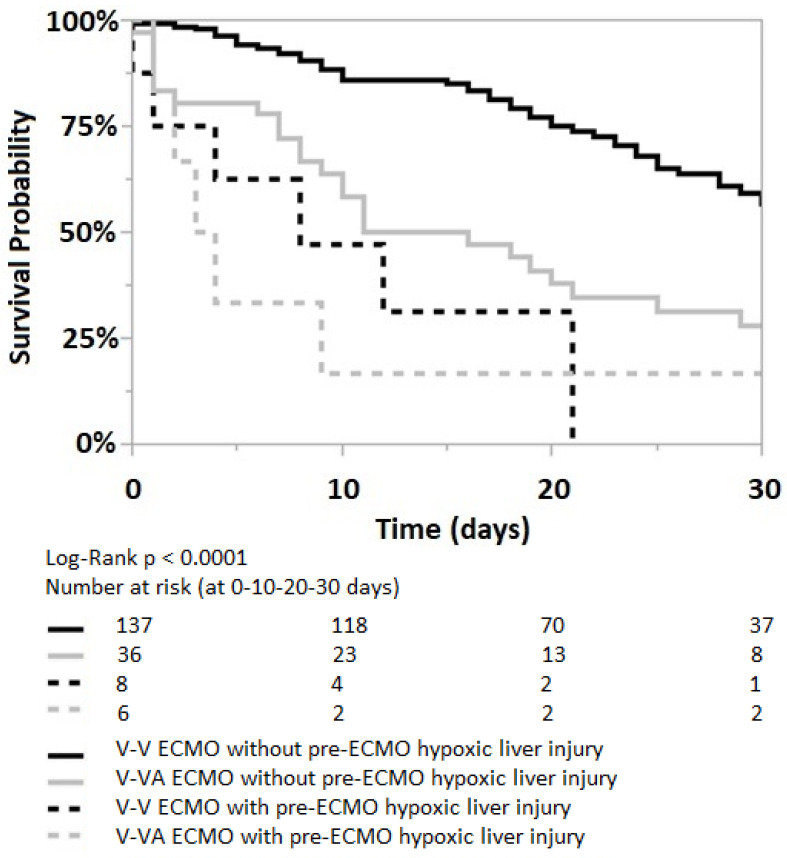
Kaplan–Meier curve for patients without (solid lines) and with (dotted lines) pre-ECMO acute/hypoxic liver injury; both in V-V and V-VA ECMO groups (black and grey lines, respectively). Log-Rank *p* < 0.0001.

**Figure 4 jcm-12-04860-f004:**
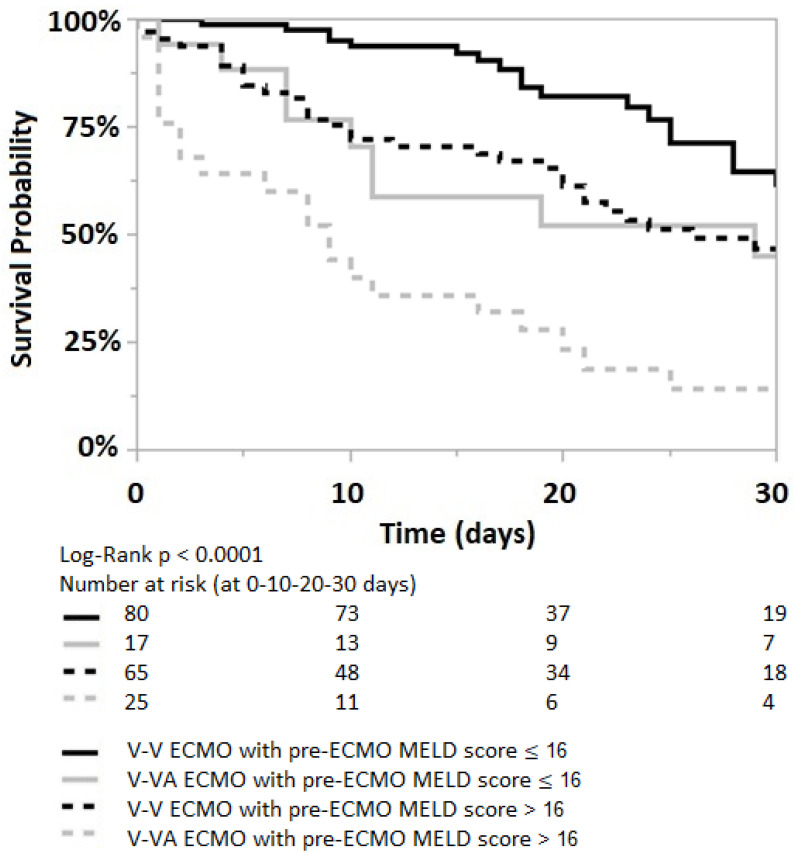
Kaplan–Meier curve for patients with pre-ECMO MELD score ≤ 16 (solid lines) and pre-ECMO MELD score > 16 (dotted lines); both in V-V and V-VA ECMO groups (black and grey lines, respectively). Log-Rank *p* < 0.0001.

**Table 1 jcm-12-04860-t001:** Pre-ECMO patient’s demographics and characteristics (survivor vs. nonsurvivor).

	Survivors n = 95	Nonsurvivors n = 92	*p* Values
Age (years)	55 (42–61)	57 (49–64)	0.07
Sex	female n = 29 (31%) male n = 66 (69%)	female n = 30 (33%) male n = 62 (67%)	0.87
Body-mass index (kg/m^2^)	29 (25–35)	27 (25–31)	0.08
ICU length of stay (days)	21 (14–33)	11.5 (6–24)	<0.0001
ECMO strategies			
V-V ECMO	n = 84 (88%)	n = 61 (66%)	0.0004
V-VA ECMO	n = 11 (12%)	n = 31 (34%)	
Duration of ECMO support (days)	12 (8–16)	9.5 (4–19)	0.08
Clinical presentation prior to ECMO initiation other than respiratory failure:			
chronic liver disease	n = 1 (1%)	n = 4 (4%)	0.2
chronic renal disease	n = 3 (3%)	n = 9 (4.5%)	0.08
cardiac failure	n = 18 (19%)	n = 40 (43%)	0.0005
septic shock	n = 53 (56%)	n = 62 (67%)	0.13
acute liver injury	n = 3 (3%)	n = 11 (12%)	0.03
Bilirubin	0.6 (0.3–1.2)	0.9 (0.5–1.8)	0.01
Aspartate transaminase	77 (38.2–146.5)	143 (58.2–414)	0.0002
Alanine transaminase	39 (28–70.2)	53 (30–159)	0.02
Creatinine	1.4 (0.7–2.4)	1.8 (1.1–2.9)	0.01
INR	1.1 (1.0–1.2)	1.2 (1.1–1.5)	<0.0001
MELD score	12 (8–20)	19 (11–23)	0.0004
SOFA score	13 (11–16)	15 (13–17.7)	0.001
PRESERVE score	3 (2–5)	4 (3–6)	0.005
RESP score	1 (−2–3)	0 (−2–2)	0.04
SAPS II score at ICU admission	69 (59–80)	78 (64–90)	0.002
Predicted mortality based on median SAPS II score	82.6%	91.2%	

ICU: intensive care unit; ECMO: extracorporeal membrane oxygenation; V-V: veno-venous; V-VA: veno-veno-arterial; INR: International Normalized Ratio; MELD: Model for End-Stage Liver Disease; SOFA: Sequential Organ Failure Assessment; PRESERVE: PRedicting dEath for SEvere ARDS on V-V ECMO; RESP: Respiratory ECMO Survival Prediction; SAPS II: Simplified Acute Physiology Score II.

**Table 2 jcm-12-04860-t002:** The ability of pre-ECMO risk factors in predicting ICU mortality. SAPS II is calculated at ICU admission; pre-ECMO values are assessed just before ECMO initiation.

Risk Factors	Cut-Off Values	*p*-Values (Univariate)	AUROC	*p*-Values (Multivariable)
Age	60	0.06	0.58	
Male sex		0.9		
Body-Mass Index	27.7	0.4	0.60	
ECMO type (V-V or V-VA)		0.0003		0.2
Cardiac failure		0.0003		0.4
Septic shock		0.1		
Chronic liver disease		0.2		
Chronic renal disease		0.06		
Acute liver injury		0.03		0.2
Bilirubin	0.63	0.03	0.60	
Aspartate transaminase	112	0.0008	0.66	
Alanine transaminase	109	0.02	0.6	
Creatinine	1.6	0.23	0.60	
INR	1.15	<0.0001	0.69	
MELD score	16	0.0001	0.65	0.04
SOFA score	13	0.001	0.64	0.6
PRESERVE score	4	0.009	0.61	0.06
RESP score	2	0.05	0.58	0.7
SAPS II at admission	75	0.002	0.63	0.09

AUROC: Area Under the Receiver Operating Characteristic Curve; INR: International Normalized Ratio; ECMO: extracorporeal membrane oxygenation; MELD: Model for End-Stage Liver Disease; V-V: veno-venous; V-VA: veno-veno-arterial; SAPS II: Simplified Acute Physiology Score II; SOFA: Sequential Organ Failure Assessment; PRESERVE: PRedicting dEath for SEvere ARDS on V-V ECMO; RESP: Respiratory ECMO Survival Prediction.

**Table 3 jcm-12-04860-t003:** Cox proportional hazard analyses of risk factors associated with ICU mortality. * during the first five days on ECMO support.

Risk Factors	Hazard Ratios (95% CI)	*p* Values
Pre-ECMO acute liver injury		
all patients	4.5 (2.3–8.5)	<0.0001
V-V ECMO	5.4 (2.3–12.9)	0.0001
V-VA ECMO	2.4 (0.9–6.3)	0.07
Acute liver injury during ECMO *		
all patients	4.7 (2.9–7.6)	<0.0001
V-V ECMO	5.7 (2.9–11.2)	<0.0001
V-VA ECMO	2.7 (1.3–5.8)	0.01
Pre-ECMO MELD score > 16		
all patients	1.9 (1.3–3.0)	0.002
V-V ECMO	1.7 (1.0–2.8)	0.04
V-VA ECMO	2.6 (1.2–5.6)	0.01
SAPS II > 75		
all patients	2.3 (1.5–3.5)	0.0001
V-V ECMO	1.9 (1.1–3.1)	0.01
V-VA ECMO	4 (1.8–8.6)	0.0004

V-V: veno-venous; V-VA: veno-veno-arterial; ECMO: extracorporeal membrane oxygenation; CI: confidence interval.

## Data Availability

The analyzed datasets for this study are available from the corresponding author upon reasonable request.

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
