# Peer review of "The Outcome Relevance of Pre-ECMO Liver Impairment in Adults with Acute Respiratory Distress Syndrome"

_jcm, 2023, doi:10.3390/jcm12144860_

Round 1

Reviewer 1 Report

First, I want to congratulate authors for their great job.

I find this work interesting, since despite the existence of some precedent evaluating this aspect in similar patients, there are doubts about its potential usefulness.

In this case, a score designed to assess chronic end-stage liver failure is used to predict mortality in patients with severe ARDS managed with ECMO.

The hypothesis is interesting and has been corroborated, however, I have certain comments for the authors to consider in order to present a more solid work and corroborate the true usefulness of the application of the MELD to predict mortality in this patient population.

First. The MELD consists of 3 parameters (Creatinine, Bilirubin and INR), this could be an analogue of other severity assessment scales, such as the SOFA, although it does not assess other extremely important aspects involved in severity, co-morbidity and mortality in these patients, such as cardiovascular failure. My perception is that the MELD assesses variables that are a reflection of severity or secondary impairment by the mechanisms that have been discussed by the authors, although it has not been assessed comprehensively by leaving out other variables that in my opinion are more important. In relevant clinical trials in patients with ARDS (PROSEVA) and even with ECMO (EOLIA; CESAR), it has been found that the main cause of death is cardiovascular failure (shock and sepsis), and in view of the fact that the MELD has not been evaluated in comparison with other similar scales such as the SOFA, or other specific scales applied in patients with V-V ECMO and ARDS, such as the RESP Score or PRESERVE, it cannot be known whether there is a real advantage in the applicability of the MELD, in the context of patients with ARDS and V-V ECMO.

Therefore, II recommend authors to extend it efforts and perform a comparative analysis of performance of MELD vs other severity and mortality predictions scales as SOFA, RESP and PRESERVE, and perhaps to consider a new applicable scale combining previously mentioned scores on V-V ECMO with MELD.  Also, deepen the literature search for relevant publications where aspects related to liver failure in ECMO patients have been studied, its role in morbidity and mortality, but with potential recovery in survivors. Also, other works to better support and describe certain pathophysiological aspects in patients with Hybrid ECMO (VV-A or VAV), which could explain the findings in the presented work.

Author Response

July 13th, 2023

Prof. Dr. Emmanuel Andrés

Editor-in-Chief

Prof. Dr. Michael G. Hennerici

Editor-in-Chief

Journal of Clinical Medicine

Ref: JCM-2406839-R1: The outcome relevance of pre-ECMO liver impairment in adults with acute respiratory distress syndrome

Dear Prof. Andrés, dear Prof. Hennerici, dear Mrs. Strugariu

We would like to thank you for the opportunity to submit a revised version of our manuscript entitled: “The outcome relevance of pre-ECMO liver impairment in adults with acute respiratory distress syndrome” to the Journal of Clinical Medicine. The manuscript has been revised, restructured, and rewritten according to yours and the Reviewers’ suggestions. We also confirm the open-review status of our manuscript. We hope the revised manuscript can be considered for publication in the Journal of Clinical Medicine.

Sincerely yours,

Stany Sandrio, on behalf of the Co-Authors

Dept. of Anaesthesiology and Intensive Care Medicine

University Medical Center Mannheim

Medical Faculty Mannheim of the Heidelberg University

Theodor-Kutzer-Ufer 1-3, 68167 Mannheim

Mail: Stany.Sandrio@umm.de

Editor

Editor comment #1: The main text of the manuscript is quite brief, which may mean that the experiment, research background, future research directions, or possible applications of the research are not described in enough detail.

Response: We would like to thank the Editor for this comment. This concise paper aims to assess the significance of pre-ECMO liver dysfunction on patient outcomes. To enhance the clarity of the manuscript, several paragraphs have been added or rephrased.

RELATED REVISED MANUSCRIPT TEXT (or Table/Figure): Introduction, Material and Methods.

MANUSCRIPT LOCATION: Lines 34-50, 58-68, 91-116, 128-148.

Editor comment #2: There are some highly repeated paragraphs or sentences in your manuscript.

Response: We thank the Editor to point this out. The repeated paragraphs are mainly located in the appendix A, which depict our ECMO workflow and management procedures. We have chosen to exclude these paragraphs from the appendix and instead provide a concise summary in the materials and methods section under "ECMO Management." Additionally, we reference our previous paper for further details on this aspect [1]. We also rephrased the repetitive sentences in the results section.

RELATED REVISED MANUSCRIPT TEXT (or Table/Figure): Materials and Methods, Results.

MANUSCRIPT LOCATION: Lines 101-116, 264-268, 275-277, 292-294 and 306-314.

Reviewer 1

Reviewer # 1 comment # 1: In relevant clinical trials in patients with ARDS (PROSEVA) and even with ECMO (EOLIA; CESAR), it has been found that the main cause of death is cardiovascular failure (shock and sepsis).

RESPONSE: We agree with the Reviewer on this aspect. Indeed, cardiovascular failure plays a crucial role in ARDS, significantly influencing our ECMO cannulation strategy. In our center, veno-venous (V-V) ECMO is utilized to support patients with isolated respiratory failure. However, in cases where patients exhibit primary respiratory failure accompanied by hemodynamic instability caused by acute cor pulmonale or catecholamine-refractory septic shock despite preload optimization as well as vasopressor and inotropic support, we initiate veno-venoarterial (V-VA) ECMO. We clarified this important issue in the introduction as well as materials and methods section under "ECMO Management). To effectively guide our cannulation strategy (V-V vs V-VA ECMO), the vasoactive inotropic score at ECMO initiation, as previously discussed in our earlier report [1], proves to be an invaluable tool.

RELATED REVISED MANUSCRIPT TEXT (or Table/Figure): Introduction, Material and Methods.

MANUSCRIPT LOCATION: Lines 34-38, 63-68, 107-116.

Reviewer # 1 comment # 2: I recommend authors to perform a comparative analysis of performance of MELD vs other severity and mortality predictions scales as SOFA, RESP and PRESERVE, and perhaps to consider a new applicable scale combining previously mentioned scores on V-V ECMO with MELD.

RESPONSE: We thank the reviewer for this valuable comment. In accordance with your feedback, we conducted an evaluation of SOFA (Sequential Organ Failure Assessment), a well-established scoring system for predicting sepsis-associated mortality and ICU outcomes. Furthermore, we examined specific respiratory ECMO scores, namely RESP (Respiratory ECMO Survival Prediction) and PRESERVE (PRedicting dEath for SEvere ARDS on V-V ECMO) scores. These aspects have been incorporated into the abstract, materials and methods, results and discussion sections of our paper. The SOFA score encompasses a broader range of variables and offers a comprehensive representation of organ system dysfunction compared to the MELD score. However, when combining the SOFA and MELD scores, redundancy arises with respect to creatinine and bilirubin evaluation. Furthermore, in our cohort, we observed that the SOFA score demonstrated a weaker association with mortality. Consequently, we conclude that a combination of the SOFA and MELD scores would not provide additional value in our study. The PRESERVE score also incorporates the SOFA score > 12 as part of its calculation, which renders the MELD’s evaluation of creatinine and bilirubin redundant. While the combination of pre-ECMO MELD > 16 and RESP score < 2 is also associated with mortality (p = 0.04), it is worth noting that the association is notably stronger in isolated MELD score (p = 0.0001). Thus, we do not include this result in our paper.

RELATED REVISED MANUSCRIPT TEXT (or Table/Figure): Abstract, Materials and Methods, Results, Tables 1-2 and Discussion.

MANUSCRIPT LOCATION: Lines 16-18, 142-148, 157-161, 187-190, 254-256,388-390.

Reviewer # 1 comment # 3: Provide relevant publications where aspects related to liver failure in ECMO patients have been studied, its role in morbidity and mortality, but with potential recovery in survivors.

RESPONSE: According to the recommendations of the Reviewer, we add these aspects in the introduction section.

RELATED REVISED MANUSCRIPT TEXT (or Table/Figure): Introduction.

MANUSCRIPT LOCATION: Lines 38-50.

Reviewer # 1 comment # 4: Provide other works to better support and describe certain pathophysiological aspects in patients with Hybrid ECMO (V-VA), which could explain the findings in the presented work.

RESPONSE: According to the recommendations of the Reviewer, we add these aspects in the introduction, material and methods, discussion and conclusion section.

RELATED REVISED MANUSCRIPT TEXT (or Table/Figure): Introduction, Material and Methods, Discussion and Conclusion.

MANUSCRIPT LOCATION: Lines 63-68, 107-112, 115-116, 341-376, 449-453.

Reviewer 2

Reviewer # 2 comment # 1: In line 36, "liver injury and dysfunction are commonly observed [2]". The study referred here only targeted a specific population (ARDS with H1N1) but did not declare any information about the rate of the liver injury and dysfunction. The authors need to provide another stronger case to support the statement.

RESPONSE: In response to the Reviewer's recommendations, we have rephrased the sentence and incorporated relevant literature demonstrating the significance of bilirubin and MELD score, which reflect liver function, in patients undergoing respiratory ECMO.

RELATED REVISED MANUSCRIPT TEXT (or Table/Figure): Introduction.

MANUSCRIPT LOCATION: Lines 36-50.

Reviewer # 2 comments # 2 and # 3: #2 In line 288-289, "However, it’s association with pre-ECMO acute liver injury does not reach statistical significance". This sentence doesn't make sense and need to correct the grammar mistake; and #3 To discuss the potential reason why pre-ECMO acute liver injury is associated with a 5.4 higher mortality risk in V-V ECMO group, while that of V-VA ECMO groups is statistically non-significant.

RESPONSE: In response to the Reviewer's comment, we have made significant improvements to better clarify this important issue. Our findings show that pre-ECMO acute liver injury is associated with a significantly 5.4 higher mortality risk in the V-V ECMO group. In the V-VA ECMO group however, although its pre-ECMO transaminase levels are higher compared to the V-V group, the association between pre-ECMO acute liver injury and mortality does not reach statistical significance. This observation can be attributed to the profound hemodynamic instability in conjunction with hypoxemia prior to V-VA ECMO initiation, which contributes to mortality in V-VA patients irrespective of the presence or absence of pre-ECMO liver injury. We add this aspect in the discussion section.

RELATED REVISED MANUSCRIPT TEXT (or Table/Figure): Results, Discussion.

MANUSCRIPT LOCATION: Lines 287-289, 350-359.

Reviewer # 2 comment # 4: To discuss the potential reason why the development of acute liver injury during the first five days on V-V ECMO is associated with 5.7 higher mortality; while that of V-VA ECMO is associated with 2.7 higher mortality risk, which is lower than the V-V ECMO group.

RESPONSE: We thank the reviewer for this valuable comment. The development of acute liver injury during the first five days on V-V ECMO support is associated with 5.7 higher mortality (p < 0.0001); while liver injury during V-VA ECMO is associated with 2.7 higher mortality risk (p = 0.01). Among patients with acute liver injury during ECMO, the V-V ECMO group exhibits a higher mortality risk compared to the V-VA ECMO group. While this difference could be partially attributed to the ability of V-VA ECMO to stabilize hemodynamics, ensure adequate oxygen supply, and mitigate additional end-organ damage, it does not reach statistical significance (95% CI 0.5 - 2.6, p = 0.7). We add this aspect in the discussion section under “acute liver injury”.

RELATED REVISED MANUSCRIPT TEXT (or Table/Figure): Discussion.

MANUSCRIPT LOCATION: Lines 368-376.

Reviewer # 2 comment # 5: In Appendix A, Clinical workflow and management strategy for patients on respiratory ECMO, the authors detailed the indication for V-V ECMO and V-VA ECMO. We can see that the clinical status of patients required for V-VA ECMO are worse for V-V ECMO, which therefore it's an important confounder that affect the mortality. I suggested the authors to clarify it in conclusion.

RESPONSE: We thank the Reviewer to point this out. We rephrased the conclusion section and provided a concise summary in the introduction section as well as in the material and methods to provide greater clarity regarding the severity of pre-ECMO respiratory failure and hemodynamic disturbance in the V-VA group.

RELATED REVISED MANUSCRIPT TEXT (or Table/Figure): Introduction, Material and Methods, Conclusion.

MANUSCRIPT LOCATION: Lines 63-68, 107-116 and 449-453.

References:

  1. Sandrio, S.; Krebs, J.; Leonardy, E.; Thiel, M.; Schoettler, J.J. Vasoactive Inotropic Score as a Prognostic Factor during (Cardio-) Respiratory ECMO. J Clin Med 2022, 11, doi:10.3390/jcm11092390.

Reviewer 2 Report

Thank you for the opportunity given from the journal editor to read and comment on it. There are a few comments that should be addressed by the authors:

1. In line 36, "liver injury and dysfunction are commonly observed [2]". The study referred here only targeted a specific population (ARDS with H1N1) but did not declare any information about the indication rate of the liver injury and dysfunction. The authors need to provide another stronger case to support the statement.

2. In line 288-289, "However, it’s association with pre-ECMO acute liver injury does not reach statistical significance". This sentence doesn't make sense and need to correct the grammar mistake. In addition, I suggest the authors to further discuss discuss the potential reason of the relative results:

·        Why pre-ECMO acute liver injury is associated with a 5.4 higher mortality risk in V-V ECMO group, while that of V-VA ECMO groups is statistically non-significant.

·        The development of acute liver injury during the first five days on V-V ECMO is associated with 5.7 higher mortality; while that of V-VA ECMO is associated with 2.7 higher mortality risk, which is lower than the V-V ECMO group.

3. In Appendix A, Clinical workflow and management strategy for patients on respiratory ECMO, the authors detailed the indication for V-V ECMO and V-VA ECMO. We can see that the clinical status of patients required for V-VA ECMO are worse for V-V ECMO, which therefore it's an important confounder that affect the mortality. I suggested the authors to clarify it in conclusion. 

Author Response

(The authors gave the same response as above.)

Round 2

Reviewer 1 Report

Well done.